# Task Migration and Scheduler for Mixed-Criticality Systems

**DOI:** 10.3390/s22051926

**Published:** 2022-03-01

**Authors:** Jeanseong Baik, Jaewoo Lee, Kyungtae Kang

**Affiliations:** 1Department of Computer Science and Engineering, Hanyang University, Seoul 04763, Korea; jsbaik@hanyang.ac.kr; 2Department of Industrial Security, Chung-Ang University, Seoul 06974, Korea

**Keywords:** real-time systems, vehicle safety, scheduling algorithms, mixed-criticality, task migration, system implementation, system analysis and design

## Abstract

The interference between software components is increasing in safety-critical domains, such as autonomous driving. Low-criticality (LC) tasks, such as vehicle communication, may control high-criticality (HC) tasks, such as acceleration. In such cases, the LC task should also be considered as an HC task because the HC tasks relies on the LC task. However, the difficulty in guaranteeing these LC tasks is the catastrophic cost of computing resources, the electronic control unit in the domain of vehicles, required for every task. In this paper, we theoretically and practically provide safety-guaranteed and inexpensive scheduling for LC tasks by borrowing the computational power of neighbored systems in distributed systems, obviating the need for additional hardware components. As a result, our approach extended the schedulability of LC tasks without violating the HC tasks. Based on the deadline test, the compatibility of our approach with the task-level MC scheduler was higher than that of the system-level MC scheduler, such that the task-level had all dropped LC tasks recovered while the system-level only had 25.5% recovery. Conversely, from the worst-case measurement of violated HC tasks, the HC tasks were violated by the task-level MC scheduler more often than by the system-level MC scheduler, with 70.3% and 15.4% average response time overhead, respectively. In conclusion, under the condition that the HC task ratio has lower than 47% of the overall task systems at 80% of total utilization, the task-level approach with task migration has extensively higher sustainability on LC tasks.

## 1. Introduction

Since the rise of autonomous vehicles, automotive systems have comprised multiple highly functional software components interacting with each other. Based on ISO 26262 [1], a fault in lower automotive safety-integrity-level (ASIL) components should not interfere with higher ASIL components, as it can be hazardous. Unfortunately, in advanced driver-assistance systems, an ASIL B communication component might control an ASIL D cruise-control component while braking or making speed-control decisions. In this case, the breakdown in the communication component would lead to chain failure in the cruise-control component and result in injuries or death. Owing to escalating costs, manufacturers cannot easily assign a higher ASIL to every single vehicle component to avoid this interference.

Traditionally, to prevent interference from lower ASIL components, automotive industries have designed vehicles by isolating different levels of safety-critical components, which is a system design approach called partitioned architecture [2]. Currently, this approach is becoming more flexible for two reasons. First, vehicle designs are consolidated based on the standard ISO 26262 and AUTOSAR 2.0 system design [3]. Scheduling diverse levels of safety functions in single systems are also being considered in future vehicle designs. Because of the popularity of these standards, electronic control units designing companies such as Siemens and BlackBerry Limited are considering mixed safety-criticality scheduling in their products [4,5]. Second, there is an incredible degree of interaction between the functions of different levels of ASIL components in autonomous vehicles, and it is difficult to classify the safety level of each function. In other words, the interference and dependencies between low and high ASIL components should be considered. State-of-the-art research addressed this issue using mixed-criticality (MC) scheduling theory [6]. In MC scheduling, each function is classified into different safety criticality levels while prioritizing high-criticality (HC) tasks and dropping low-criticality (LC) tasks in scheduling overhead. However, MC scheduling theories do not fully consider the interactions between the different criticality levels.

In this paper, a safety-guaranteed and inexpensive scheduling are proposed, both theoretically and practically, for LC tasks that HC tasks rely on. In MC systems, initially, LC tasks drop their missions when a scheduling overhead leads to insufficient CPU time for all tasks. This policy also applies to essential LC tasks. The important LC tasks then undergo task migration from the residing system to the external system. Upon evaluation, it was observed that the approach taken in this study is more compatible with the task-level MC scheduler compared with the system-level MC scheduler.

Figure 1 shows an example of our approach in comparison with the role-changing scenario of truck platooning. Truck platooning is a distributed system that includes one driver and multiple trucks, in which the trucks following the lead truck rely on the communication tasks of the lead truck over the system. For each following truck, the communication task controls the acceleration task; therefore, the communication tasks are considered HC tasks. However, in the lead truck, the acceleration task has no dependency on the communication task; thus, the task is considered at the LC level. In MC scheduling theory, LC tasks are dropped whenever the lead truck has computational overhead. An HC task in the lead truck has no dependency on the dropped LC task. Owing to dependencies, HC tasks relying on an LC task fail in-chain. Using our approach, the dropped LC tasks are migrated, and the safety of the distributed system is preserved.

To guarantee safety, task migration should be carefully positioned without affecting the predictability of the HC tasks. Therefore, we adopted practical task migration features that solve the concerns caused by migration latency during state transfer [7], which is one of the well-known major drawbacks. The following are the contributions that, in real-time, bind the task migration and prioritize the predictability of HC tasks while rescheduling LC tasks that were dropped by conventional MC schedulers.

### Contributions

Practical task migration usage: For important LC tasks, task migration was applied, and global deadlines were established. Compared to other resilient methods, task migration is fully sustainable; however, the massive amount of context data burdens timing constraints. However, our system design sets a limit to the time consumed by task migration and minimizes the amount of context data by restricting the active data. The timing of the LC task migration was guaranteed using globally synchronized deadlines between systems as determined by real-time guaranteeing calculations. For hardware configurations, the Wi-Fi-direct and advanced hardware modules were approximately 13.3 times faster and 6.3 times more stable than the Wi-Fi-router in task migration. Based on the results, hardware component upgrades and bandwidth reservations are suggested to guarantee better task migration timing.Theoretical analysis of using task migration: Task migration requires a thorough utilization guarantee and minimum response time in schedulability analysis and in practical application. This paper provides a theoretical analysis for scheduling task migration in worst-case MC scheduling scenarios.MC scheduler implementations on Linux Kernel: Conventional MC schedulers are dedicated to schedulability analysis and simulation-based evaluations because it is difficult to implement them in operating systems such as Linux. Fortunately, the Linux Kernel has versions with real-time patches for the earliest deadline first (EDF) schedulers and provides modifications at the source level. The EDF scheduler was updated from EDF-VD into the MC-scheduler MC-ADAPT with task migration features. Linux Kernel version 4.14.91-rt49-v7 (ver. PREEMPT_RT) was implemented with configurations on armv7l to port the OS into Raspberry Pi 3B+ with uniprocessor configurations for detailed results. For the implementation, pre-existing deadline tools, such as deadline_test and ftrace, were adopted.Evaluations of MC schedulers and our approachTask-level MC scheduler preference: From the experiments, it was observed that only one task could be handled at a time during the migration. Thus, the task-level MC scheduler is more compatible compared with the system-level MC scheduler. The results showed that 25.5% of the dropped LC tasks regenerated via the system level, while 100% regenerated via LC task utilization of 47%.Investigating HC task violations: From the deadline test, it was observed that task systems with a larger HC task ratio would fail task-level MC schedulers with a high likelihood. By implementing migration, HC task violations at the system-level and task-level were 11.3% and 69%, respectively, and the average violation response times were 15.4% and 70.3% under task systems with a higher HC task utilization. Task systems with a higher LC task utilization had nominal violation rates, less than 10%, for both system-level and task-level EDF-based MC schedulers using our approach.

Organization of this paper. The rest of this paper is organized as follows. Section 2 presents the problem statements. Section 3 provides a background on MC scheduling. Section 4 explains our distributed approach to MC systems, i.e., using task migration and guaranteeing it by timing constraints, especially at critical instants. Section 5 describes the distinct features of implementing EDF-VD and MC-ADAPT scheduler in Linux kernel using the task migration kernel module. Section 6 presents an evaluation of the task migration in MC systems. The evaluation covers the acceptance of tasks, the deadline miss ratio, the violation of HC tasks, end-to-end task migration delays determined by different connections, and scheduling overheads compared to the conventional SCHED_DEADLINE scheduler. Section 7 presents a comparison with related works, with a particular focus on other useful, resilient methods in real-time and mixed-criticality systems. Section 8 comprises some discussions on using our approach. Finally, Section 9 provides a summary of using our distributed approach in MC systems.

## 2. Problem and Motivation

The objective of this work is to guarantee safety-critical distributed systems from the perspective of real-time scheduling. Applying dynamic function theory to MC schedulers and the scheduling overhead in task migration are the challenges that need to be addressed in safety-critical distributed systems.

Runtime dependencies: The specific interest is to schedule tasks that consider dynamic critical-level changes. First, a high-to-low criticality change is always safety-guaranteed because HC tasks are already meant to be guaranteed at the time of system design. Thus, it can be considered as an overreaction to LC tasks to ensure a safety guarantee, and the task remains safety-guaranteed. Lowering critical levels may cause the same problem of overreaction, and the task is safety-guaranteed; however, there is no need to assert this because the tasks eventually become LC tasks. Second, a high-to-low dynamic change is the incremental change in critical levels, and because the tasks are scheduled by an LC-level policy, the tasks are not guaranteed. Our study focused on the dynamic incremental criticality changes in these tasks.

Theory into practice: Most theoretical studies on MC scheduling conclude their work by accepting task systems and covering various cases of task scheduling at system runtime, which are called online schedulability tests. These tests are theoretically acceptable but require verification in an actual operating system, such as the Linux Kernel.

Task migration overheads: Compared to other resilient methods, task migration can fully sustain task details and is effective in distributed systems. However, task migration methods have trade-offs between full sustainability and transfer latency, owing to the extensive memory used by the tasks. Addressing task migration in safety-critical systems with real-time guarantee, while achieving the sustainability of important tasks, is a challenging aspect of distributed safety-critical systems. In addition, cases in which the migration target task also needs a real-time guarantee need to be considered.

## 3. Mixed-Criticality Scheduling Background

This section, for ease of explanation, provides the basic background of MC scheduling for both system-level and task-level MC schedulers [8,9].

### System Model

The system model of our approach consists of MC-ADAPT [9] task features and additional specifications particular to our approach. For simplification, dual-criticality uniprocessor systems are considered for scheduling implicit-deadline sporadic tasks.

Task Model: Here, an implicit deadline sporadic task system for n MC tasks is considered. In Table 1, the description representing each MC task $\tau_{i}$ is characterized by $\left(T_{i},C_{i}^{L},C_{i}^{H},\chi_{i}\right)$. The tasks are classified according to their criticality levels into sets of HC and LC tasks. For example, a set of HC tasks are denoted as $\tau_{H}:\left\{\tau_{i}\in \tau |\chi_{i}=H\right\}$, and a set of LC tasks are denoted as $\tau_{L}:\left\{\tau_{i}\in \tau |\chi_{i}=L\right\}$.

Utilization: Because the period and worst-case execution time (WCET) are denoted as $T_{i}$ and $C_{i}^{L}$ (or $C_{i}^{H}$), respectively, each task has multiple WCETs for each criticality level and a total of two sets of WCETs for each task for dual criticality. For each task $\tau_{i}$, the utilization of high and low confidences are $u_{i}^{H}=C_{i}^{H}/T_{i}$ and $u_{i}^{L}=C_{i}^{L}/T_{i}$, respectively. The utilization of the task system, bounded by criticality levels, is expressed as follows:(1)$$U_{H}^{H}=\sum_{\tau_{i}\in \tau_{H}}u_{i}^{H},U_{H}^{L}=\sum_{\tau_{i}\in \tau_{H}}u_{i}^{L},$$
(2)$$U_{L}^{H}=\sum_{\tau_{i}\in \tau_{L}}u_{i}^{H},U_{L}^{L}=\sum_{\tau_{i}\in \tau_{L}}u_{i}^{L}.$$

Behavior Model: Figure 2 presents the behavioral model of MC tasks. Compared with conventional MC schedulers, the migrated state is extended to LC tasks [9]. An important feature of the LC task behavior is that the LC task state is returned to active from dropped. This resilient behavioral model for the LC task was enabled by exploiting another system in distributed systems.

## 4. Proposal: The Distributed Approach

A system design is proposed for distributed mixed-criticality (DMC) scheduling, which is a platform for scheduling LC tasks dropped in distributed systems. To schedule the dropped LC tasks, DMC assigns the LC tasks to another system using real-time bound task migration to guarantee the deadline of the LC task. In Figure 3, the temporal overview of DMC explains the DMC’s features and expected effects. When the HC task requires more than the given execution time, the HC task exceeds the runtime, and some LC tasks are sacrificed in conventional MC schedulers. Specifically, the LC task is dropped because the scheduled task system exceeds the CPU resource limit; however, DMC transfers the LC task to another available system and guarantees a preset deadline. From the perspective of the initial task system, the system is scheduled with more than 100% CPU utilization.

### 4.1. Positioning the DMC Task

The schedulability of DMC is analyzed using task migration as a task. First, the worst-case execution time of the migration task is measured, and then the upper bound of the migration utilization is calculated. For the proof, some preconditions, which are plausible based on the probability of the events, have to be satisfied.

Cond. 1: Only one HC task is mode-switched at an instance.Cond. 2: The WCET of task migration is smaller than the WCET of target LC task.

Initially, the system has enough resources to accept additional tasks at approximately 20% utilization owing to the marginal safety-critical system design. In addition, the mode switch of both source and destination systems in the same instance is unlikely; however, defining the critical instant of accepting the sporadic task is complex, which is the topic of sporadic MC scheduling theories [10]. Based on IEC 61508, which ISO 26262 is derived from, the safety integrity level (SIL) 4 is associated with the probability of the dangerous failure rate limit of $10^{-9}$ per hour; the concurrent failure of SIL 4 would be $10^{-9}\times 10^{-9}=10^{-18}$ per hour, which might occur once in 117 trillion years.

#### 4.1.1. Measuring Worst-Case Execution Time of Migration

The essential part of addressing migration in practice is measuring the actual task migration latency and guaranteeing the measured worst-case latency. The measuring methodology requires safety margins, which are set by the domain, usually 20% in safety-critical systems [11]. The measurement is defined by the task migration procedure, the number of target tasks, and the role of the system in migration. The migration procedure consists of dump, copy, and restoration tasks. The number of target tasks is approximately one, some, or all LC tasks, which determines the amount and complexity of data transfer through a network. The system’s role is to describe the source and destination system to determine the required migration procedures.

#### 4.1.2. Utilization Upper Bound of Migration Task

When the system is in the HC mode, the DMC task should also be scheduled as an HC task; therefore, the utilization of task migration with the EDF-VD scheduler can be expressed as follows:(3)$$xU_{L}^{L}+U_{H}^{H}+U{}_{m}\leq 1.$$
1:EDF-VDAcceptanceatLC-mode
where the coefficient *x* is a constant that determines the upper bound of the execution time of the HC tasks. The smaller this coefficient, the larger the upper bound of the HC task runtime; therefore, *x* should be as small as possible.
(4)$$\left(Because\;U_{L}^{L}+\frac{U_{H}^{L}}{x}\leq 1.\to x=\frac{U_{H}^{L}}{1-U_{L}^{L}}.\right)$$
(5)$$\to \frac{U_{H}^{L}}{1-U_{L}^{L}}\cdot U_{L}^{L}+U_{H}^{H}+U_{m}\leq 1.$$
(6)$$Therefore,\;U{}_{m}\leq 1-\frac{U_{H}^{L}\cdot U_{L}^{L}}{1-U_{L}^{L}}-U_{H}^{H}.$$
2:Upperboundutilizationfortaskmigraiton

After finding the upper bound utilization of the DMC task, the utilization should be extended to the actual execution limit to identify the feasibility of the migration task. The calculation of the upper bound utilization on the destination system is deferred based on the premise that the destination system for task migration should always be available to accept dropped LC tasks.

#### 4.1.3. Deadline Calculation of Migration Task

Utilization is the CPU usage percentage, but the calculation must be performed with the actual execution time guaranteed at the instant of the mode switch. Determining the correct way to put migration execution time into practice without violating any HC task is the main conflicting goal. After the mode switch, the HC tasks have the same criticality levels, which means that no additional criticality scheduling can be applied. With the original EDF scheduler, the tasks were prioritized based on the earliest deadline. To satisfy the aforementioned conditions, the migration task should be scheduled as soon as possible. Therefore, we analyzed whether the migration task violates the HC task with the earliest deadline; granted a violation, the HC task with the next earliest deadline was analyzed. The process continued while comparing the tasks in ascending order until a task that was not violated by the migration task was found. At this point, a deadline earlier than that of this HC task was assigned. Because of real-time synchronization with another system, splitting the migration task multiple times is not advisable.

The lower bound of the deadline of the DMC task is calculated using the implicit deadline tasks (D=T). This condition is already satisfied because LC tasks would have been dropped, and the HC tasks would have abandoned their virtual deadlines and restored the given deadline, which is equal to the period:(7)U=C/D→∴D=C/U.

The deadline (*D*) can now be derived using the worst-case execution time (*C*) and utilization (*U*). After positioning the migration task, the interference from HC tasks with higher priorities is calculated.

### 4.2. Task System Example

Table 2 presents an example task system that satisfies the condition of EDF-VD schedulability. There is a total of 10 MC tasks, and the ratio for utilization of the criticality ($U_{H}^{L}$:$U_{L}^{L}$) is 34.4%:46.7%. At mode-switch, the utilization of HC tasks is raised to 68.7%. The first step is to measure the worst-case execution time of the migration task. Supposing that the total migration latency of the safety margin is 1 ms, the second step is to calculate the upper bound of the migration task using Equation (Equation 6), as follows:(8)$$U_{m}=1-\frac{U_{H}^{L}\cdot U_{L}^{L}}{1-U_{L}^{L}}-U_{H}^{H}=1.2\%.$$

Therefore, the utilization upper bound of the DMC task was 1.2%. This indicates that there is a possibility of the DMC task being executed when the utilization upper bound is more than 0. Considering that there is approximately 20% slack time in the LC mode and approximately 30% slack time in the HC mode, the derived upper bound seems pessimistic. The final step is to derive the shortest deadline from the previous results:(9)$$D=C/U\to D_{m}=C_{m}/U_{m}=81\;\mathrm{ms}.$$

After the final step, the HC tasks with the earliest deadlines were compared. From the results, it was verified that the DMC task has a priority greater than $\tau_{3}$ and less than $\tau_{2}$. Therefore, the DMC task is affected by $\tau_{1}$ and $\tau_{2}$. Figure 4 presents the Gantt chart of the task system example. After the mode switch, the DMC task is scheduled after $\tau_{1}$ and $\tau_{2}$. The DMC task does not violate the schedulability of HC tasks. Unfortunately, the LC tasks $\tau_{6}$ cannot satisfy the deadline, as determined by the source system.

### 4.3. Adaptation to Task-Level

A fatal issue of using the combination of a system-level MC scheduler and task migration is that a maximum number of LC tasks are dropped, and only a single task can be migrated at a time. To address this issue, the system-level MC scheduler is differentiated into a task-level MC scheduler. Conventionally, there are many task-level MC schedulers, such as MC-ADAPT [9] and AMC [12]. These MC schedulers are adaptive because of the consecutive LC task drop feature. In the system-level MC scheduler, all LC tasks are dropped, but task migration can handle only a certain number. The solution at the system level is limited, and task migration has an obvious sequential weakness because only a single task can be migrated at a time. By contrast, by dividing the variance of the mode switch of the system level into a task-level MC scheduler, a single task is dropped and migrated. This approach optimally increases the schedulability of LC tasks.

*Example.* Given the same task system presented in Table 2, the result of $U_{m}^{s}$ using the MC-ADAPT algorithm is 6.1%. This is higher than the capability of EDF-VD, which is 1.2%. The offline schedulability analysis of MC-ADAPT is complicated because of conditions in the preprocessing phase. Proving the upper bound of task migration by EDF-VD is sufficient for MC-ADAPT in the worst-case scenario that all LC tasks are dropped when all HC tasks have the same fraction of $C_{H}/C_{L}$ which we provide proofs in Appendix A.

## 5. Implementation

This section describes the overall architecture of task migration in a Linux MC system. As a preliminary, to enable MC scheduling in Linux, we converted the existing EDF scheduler into a system-level MC scheduler, which is EDF-VD, and into a task-level MC scheduler, which represents the scheduling policy in MC-ADAPT. In this MC scheduling environment, the proposed task migration is implemented independently as a kernel module interacting with MC schedulers. To simplify comparisons between schedulers and strictly observe the dropped state of LC tasks, the implementation on Linux was built to run in a uni-processor environment.

### 5.1. Conventional EDF Scheduler in Linux

SCHED_DEADLINE is a CPU scheduler based on the earliest deadline first (EDF) scheduler and constant bandwidth server (CBS) algorithms [13]. After consuming all its given execution time, the task suspends itself and reactivates close to the current deadline; however, the scheduler is designed to delay the execution of tasks with other deadlines as long as its budget allows. Such scheduling behaviors allow a soft real-time bound on tasks, introduce unpredictable deadline guarantees on tasks, and are especially fatal to HC tasks. To prevent unpredictability, we implemented hard real-time features that guarantee even the worst-case scenario of exceeding the total CPU utilization by adopting MC scheduling and task migration.

Scheduling Interface: The SCHED_DEADLINE has a simple interface, which can assign any desired scheduling policy and any schedule attribute to the created task [14]. The task uses sched_setattr() and sched_getattr() to set and read the scheduling attribute, respectively. However, the task calls the sched_yield() function to end a single instance to notify the kernel. The kernel cannot decide whether the task has finished running the single instance; therefore, the task should alert the kernel when it completes the single instance (or job) within the allotted period. The sched_yield() function causes the notifying thread to relinquish the CPU; thereafter, the thread is moved to the end of the queue for static priority, and a new thread is executed.

### 5.2. Implementation of EDF-VD Scheduler

In the EDF-VD scheduling policy, HC tasks are prioritized compared to LC tasks by providing a virtual deadline (VD) calculated by scaling down for earlier deadlines. In the original proposal of Vestal [6], the worst-case execution time for multiple levels is used. To determine the multiple levels of the tasks, a member variable crit_level is added to present corresponding multiple execution times for MC tasks.

In lines 3–24 of Algorithm 1, representing the preprocessing phase, the function __setparam_dl() is called, which is the actual function that fills the real-time and MC member variables of the scheduling entity struct sched_dl_entity. This function performs an initial build for the EDF-VD scheduler by calculating the overall utilization and deriving the least value of the coefficient *x*. If the scheduled MC task has crit_level HI, the relative deadline dl_deadline is set by the virtual deadline denoted in EDF-VD while preserving the original deadline in variable init_deadline. For the mode switch, the original deadline is restored; otherwise, if the crit_level is an LC, the attribute is set to the original deadline of the tasks, and rb_node is inserted into the task drop list, which carries all lists associated with the LC task.
**Algorithm 1:** Linux EDF-VD Scheduler.
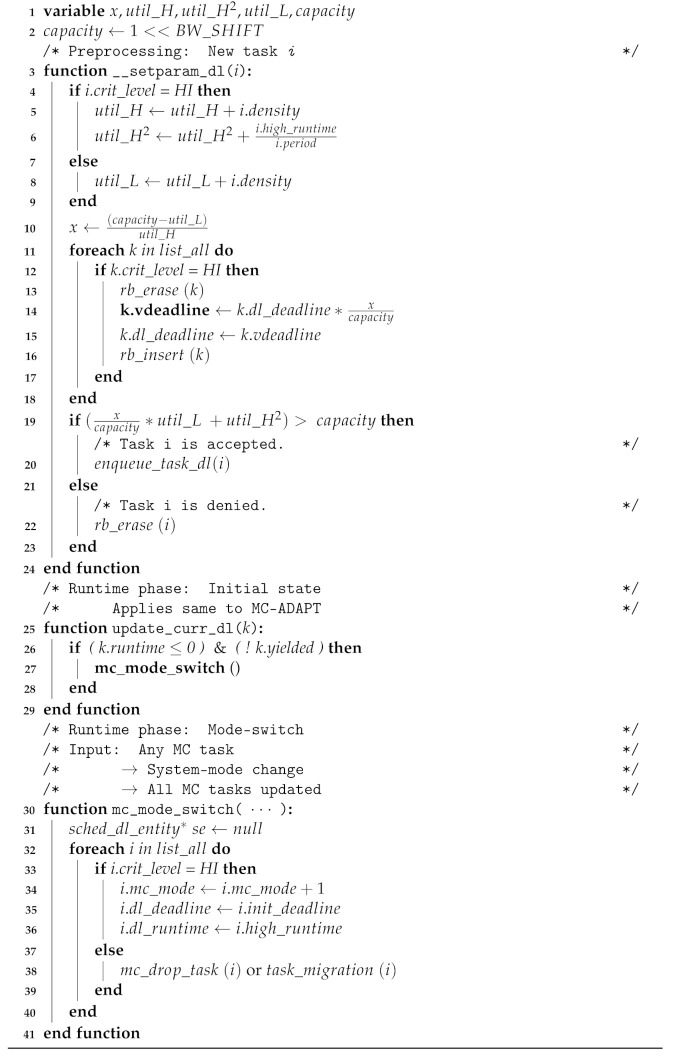


In lines 25–41 of Algorithm 1, representing the runtime phase, the occurrence of a runtime exceeding the MC tasks is monitored by the existing function update_curr_dl(). Mode switching occurs whenever the monitored HC task uses the defined execution time without finishing the job. In lines 30–41, for all HC tasks, the relative deadline is restored by the initially defined implicit deadline init_deadline instead of the virtual deadline, and the runtime is expanded by a predefined HC-level runtime high_runtime. The task drop list is used to drop the indicated LC tasks from the LC task maintaining list at one go. After these steps, there is still sufficient CPU time for mode-switched HC tasks.

Table 3 presents the worst-case time complexity with each operation cost. A denotable feature is a preprocessing phase having polynomial time complexity $O(n^{2})$, which is due to sequential task invocation at system runtime.

### 5.3. Implementation of MC-ADAPT Scheduler

The fatality at the system level is that the task level MC scheduler seems to be more efficient. The expectation of combining the task migration solution with a task-level MC scheduler is larger than that of the system-level MC scheduler because migration techniques have limitations in dealing with multiple tasks simultaneously. Task migration may exactly highlight the fewest number of tasks to drop by the task-level MC scheduler, such as MC-ADAPT, with a fairly high probability. The implementation of MC-ADAPT differs from EDF-VD in mode-switch targets, lowering the number of tasks to drop to the fewest possible.

The main update from the system-level scheduler in the preprocessing phase is the inclusion of LC tasks in the task drop list in descending order of the task utilization, as presented in lines 17–23 of Algorithm 2. This is one feature specified in MC-ADAPT to immediately address the HC mode of the HC task. In the runtime phase of MC-ADAPT, the mc_mode_switch() is generated with a single HC task rather than the entire set of HC tasks in the system-level MC scheduler. The updated part is listed in lines 36–51 of Algorithm 2. This adaptively affects the runtime. The HC tasks exceeding the runtime are measured by the system’s overall exceeded utilization, and the scheduler dumps only the LC tasks from the task drop list corresponding to the exceeded utilization.

Table 4 presents the worst-case time complexity with each operation costs. In the preprocessing phase, represented in lines 17–32 of Algorithm 2, an additional operation for building the task drop list and acceptance test is introduced. Although the time complexity remains the same as EDF-VD, which is $O(n^{2})$, the operational overhead at the preprocessing phase is predictable and addressable. Furthermore, the task-level approach provides better response time in handling mode-switch situations. The worst-case time complexity is deemed identical to the system-level approach, but the loop has a conditional break that completes the operation by the amount of exceeded utilization of one mode-switched HC task, having pseudo-polynomial time complexity (lines 40, 45–51). Typically, this on-demand approach is given as constant time complexity. Additionally, the complexity corresponds to the drop rate of LC tasks and minimizes the deadline miss ratio as a final outcome.

**Algorithm 2:** Linux MC-ADAPT Scheduler.

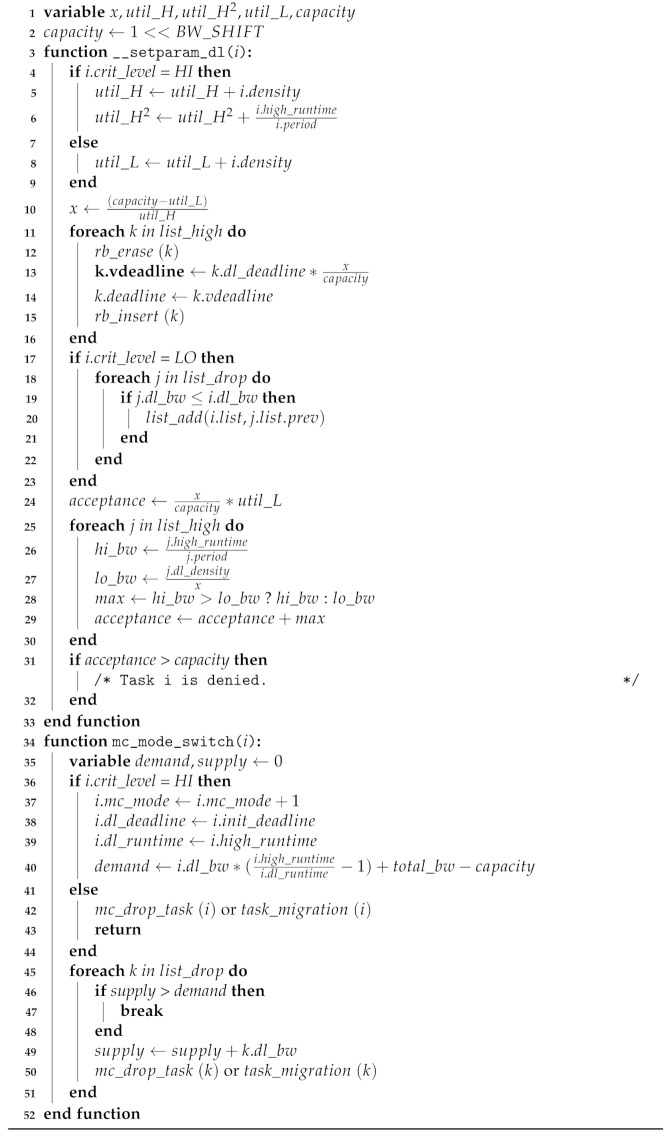



### 5.4. Implementation of DMC Kernel Module

The proposed DMC is a supplementary kernel module that can be applied to diverse MC schedulers, as shown in Figure 5. DMC was implemented using the former implementations of EDF-VD and MC-ADAPT. The main idea of utilizing the task migration kernel module is to gain more CPU time by using external systems. The LC task is displaced from the source system, but using the kernel module first dumps the memory of the LC task and transfers it to a destination system as a network packet. In the destination system, the initial form of the LC task is forked. The memory transferred from the source system is restored to the forked LC task in the destination system. In this manner, the method can obtain the CPU time from the destination system.

To introduce the DMC structure, a more specific implementation of the DMC, based on DMC strategies, is introduced. To satisfy the need of Kernel authorities and for the protection of the Kernel, the DMC is programmed in a Kernel-level module. The Kernel-level module is built with a TCP client and server model, with an interaction with the SCHED_DEADLINE source of the Linux Kernel. The DMC is designed to bind with any MC scheduler that can pass the MC packet formats when invoked.

First Stage: Scheduling MC Tasks. MC tasks are known to vary in criticality levels for each task; it is essential to set a task system with multiple criticality levels in the source system. In this stage, the tasks should be initially set to the LC mode state regardless of the criticality level; in addition, in the destination system, a dummy task is forked with a low criticality level to prepare an immediate update after the task migration. Forking a dummy task is for ledging the migration latency; however, it is not an essential part of the DMC.

Second Stage: Mode-switch. Initially, the HC tasks are scheduled with virtual deadlines, similarly for all tasks in the LC mode. Whenever any HC task exceeds the runtime, the scheduler switches the task mode to the HC mode with the function mc_mode_switch(). In the task-level approach, the amount of the LC task utilization is calculated when it exceeds 100% CPU utilization. The dropped number of LC tasks is calculated using the MC-ADAPT theory. At the mode-switch, in the worst-case, the LC task already performed is scaled to *x*, and the remaining amount is scaled to (1−x). If the dropped LC task has an undroppable flag, the LC task is pushed to the DMC kernel module waiting for migration.

Third Stage: Task Migration. A packet argument that complies with the DMC customized format is sent by the scheduler. After receiving the customized format from the scheduler, the DMC module itself should invoke an MC task with kthread_run() to perform task migration over time. The invoked Kernel thread reads the memory of the dropped LC task and packs the information into a TCP packet. The TCP packet contains a packet called an MC packet with MC features in the header. The TCP server in the destination system receives and unpacks the packet; immediately thereafter, it restores the MC and memory information into a dummy task. The dummy task can be forked at the time the packet is received, but the latency increases. The functional part of the restoration of the LC task is completed.

Last Stage: Evaluation of The LC task. The migration is completed prior to the last stage, but the LC task is continuously using information from the source system. The last stage is to check if the absolute deadline set by the source system is satisfied. This deadline satisfaction is an important part indicating that the LC task has globally performed the job seamlessly.

## 6. Evaluation

For extensive evaluation, a more specific subset defined in MC-ADAPT was used to investigate testing on five different schedulers, namely SCHED_DEADLINE, EDF-VD, MC-ADAPT, EDF-VD with DMC, and MC-ADAPT with DMC, under the same conditions. The subset was varied in the task-level mode switch with the conjunction of the criticality level of the tasks themselves. Task-level evaluation is effective because it implies a system-level evaluation. The mode-switch test was triggered by injecting more than twice the number of jobs for a task in the subset $U_{H2}$ after some time had elapsed from the beginning of the test. The subset $U_{L2}$ can be expected from the task selection policy of MC-ADAPT. The test was designed to show that the MC schedulers and the DMC module do not violate the HC tasks and that the performances of system-level and task-level MC schedulers differ.

The proposed DMC should also be thoroughly verified by the conventional deadline test used in the SCHED_DEADLINE Linux Kernel. The deadline miss ratio (DMR) test is suitable for evaluating the SCHED_DEADLINE Linux Kernel, but because the number of deadline misses is a chain reaction of the first deadline miss of a task, the test should measure if a deadline miss has occurred (deadline miss more than once or no deadline misses) to achieve our goal of measuring the tasks dropped. Hence, a DMR test was conducted to verify the MC schedulability of each Linux real-time scheduler.

The DMC-patched scheduler works as intended with proven dependability at the Linux Kernel level usage in Raspberry Pi 3B+. In this subsection, we describe the measurement of the scheduling overhead and compare the result to that of the existing SCHED_DEADLINE scheduler.

In summary, we conducted these tests because, first, it is necessary to evaluate the quality of our MC Linux scheduler implementations in a traditional way, such as acceptance ratio, deadline miss ratio, and the scheduling overhead. In the context of MC theories, the HC tasks should not be violated in any case but require observation in an excessive test. Last but not least, some conditions need to be satisfied using the task migration methods such as timing synchronizations between the distributed systems and timing predictability through the network and wireless networks in the case of mobility systems such as autonomous vehicles.

### 6.1. Acceptance in Pre-Processing Phase

To investigate the suitable HC task ratio for task migration, the schedulers were tested using task systems varying in total utilization according to the ratio of HC tasks. Each task had a high-criticality runtime that was twice or a random number of times that of the low-criticality runtime. All tasks with low-criticality runtimes were set to 10 ms, and the periods were set randomly. The task system had 20 tasks, and the test was performed more than 300 times for each case. In Figure 6a, the results of the system-level MC scheduler (EDF-VD) are displayed. The results are compared with the task-level MC scheduler (MC-ADAPT), which is shown in Figure 6b. The acceptance results are similar when the high criticality runtime is fixed to twice that of the low criticality runtime. Both results approximately have a maximum HC ratio of 0.5 on 0.8 total utilization.

Testing the task-level MC scheduler using a task system having a random high-criticality runtime sacrificed the acceptance of task systems. In Figure 6c, the maximum HC ratio is approximately 0.4 for a total utilization of 0.8, which is an approximately 20% HC ratio loss compared to the fixed conditions with a 0.5 HC ratio acceptance. The main difference between theory and practice is in the scheduling order. In practice, the task system handles each task using a sporadic scheduling policy; therefore, it is necessary to rerun and observe the acceptance test for the implemented Linux schedulers.

### 6.2. Deadline Miss Ratio

An existing test tool, called deadline_test, was modified by adding MC attributes. The time mode switch was triggered 300 times for each scheduler. As depicted in Figure 7, five real-time MC schedulers were evaluated using the DMR test. To represent the comparable behavior of each scheduler, we categorized the tasks into four main groups and put them into an affected order starting from the mode-switch triggered HC-task group HI2. These are the explanations for the four task groups:HI2: HC tasks exceeded their given LC-WCET and changed to HC-mode;HI1: HC tasks sustaining their given LC-WCET and still in LC-mode;LO2: LC tasks dropped by the mode-switched HC tasks (HI2) decisions;LO1: LC tasks remaining schedulable after mode-switch only if the scheduling policies support maintaining the tasks (e.g., MC-ADAPT).

In Figure 7, the conventional SCHED_DEADELINE scheduler (noted as EDF+CBS) dropped the HC task, which was triggered to overdo the initially given LC-WCET and implies that no criticality levels were considered and eventually leads to the worst-case scenario. None of the MC schedulers violated the high-level tasks, and all the MC schedulers performed their jobs in case of mode-switch. For the LC tasks, because the SCHED_DEADELINE scheduler already dropped the forced HC task, there were no more tasks to drop; however, the MC schedulers varied in the drop ratio of the LC tasks. According to the task dropping policy of EDF-VD, all LC tasks are dropped, while MC-ADAPT drops only 25.53%. Regarding the DMC module for each MC scheduler, the drop ratio was improved by 25.53%. The total drop rate of MC-ADAPT was eliminated from the results using the DMC task. The most important aspect is to observe whether the undroppable task has been dropped. Undroppable tasks can be defined as a combination of HC tasks and escalated LC tasks. As observed, the escalated LC task was always dropped without the DMC module, while no tasks were dropped with the DMC module. Through the deadline test, it was demonstrated that the compatibility with the task-level MC scheduler was higher than the system-level MC scheduler. The task-level had O(1), while the system-level had O(n) deadline miss ratios when tested on *n* tasks.

#### HC Task Violation Test

Figure 8a shows the execution load of each real-time scheduler. The scheduling overhead after applying the MC scheduling patch and the DMC patch were compared on LO-mode HC tasks, while other tasks were dropped in at least one scheduler. Complicated task-level MC schedulers, such as MC-ADAPT, seem to have no more overhead than using the system-level scheduler EDF-VD. However, HI-mode LC tasks seem to have a 25.4% scheduling overhead in MC-ADAPT compared to the original EDF+CBS scheduler. This overhead is tolerable because the deadline is satisfied during the test period.

Figure 8b shows the testing result using this HC superior task system, where it can be observed that the existing EDF scheduler also incurs a large overhead. This is regarded as an unstable operation as the number of tasks tested is the same, but the gap between the testing cycles is larger. First, similar to previous results, the existing EDF scheduler is significantly affected by the scheduling order. In particular, LC tasks are significantly affected, but another LC task, which has a longer cycle, is directly affected and reversed. In contrast to existing EDF, EDF-VD is very stable and can obtain scheduling results within the expected range; however, with MC-ADAPT, the HC task which should have been in LO-mode has been severely violated. Theoretically, because MC-ADAPT initially ran for a set of unacceptable tasks, it can be viewed as a failure in the acceptance test. In particular, MC-ADAPT had a smaller acceptance ratio than EDF-VD owing to the implementation. In conclusion, because the DMC module is eventually implemented as a module independent of the scheduler, the schedulers are not affected by the supplementary DMC module.

The results of the violation test suggest that schedulability analysis at the design time is not sufficient to guarantee the actual schedulability of HC tasks. HC superior task system configurations should be avoided by both system-level and task-level MC schedulers; in particular, they are fatal for task-level MC schedulers.

### 6.3. Global Deadline with Real-Time Guarantee

To guarantee the migrated LC task in other systems, the preset deadline must be transformed into a global time between the systems. The transformation of the LC task deadline into a global deadline is shown in Figure 9. In the Linux Kernel, the deadline schedulers handle the deadlines in their own system time, which may differ from the system boot time, known to rq_clock(). To address this time gap between systems, the time of the system background was eliminated, leaving the remaining time until the deadline. The measured network latency was then decreased by a safety margin of 20%, which provided the worst-case remaining time for LC task execution. Both the deadline and remaining runtime should be considered; however, the runtime of the task is independent of the system time and is considered to be the same as the global runtime between the systems. In this study, the focus was on the real-time bound of the utilization and synchronization methods. For reference, our migration features are displayed in Table 5. Network latency was measured using the ping command. From the results, it can be observed that using WiFi-direct provides a more stable network latency than the router, which enables bandwidth reservation only for connected systems. It is also better to use a combination of WiFi-direct and a network module. In our experiment, the NEXT-1302WBTA network module was used, resulting in a 1.8 times faster communication on average. In total, network responsiveness highly depends on the specifications of the hardware devices and bandwidth reservations. Real-time bounds can be extended through network bandwidth. Some previous studies bind network latency, such as TTEthernet [15], to TSN [16].

### 6.4. Scheduling Overhead

To analyze the latencies and performance of real-time related functions, the Linux Kernel function-tracing interface called Ftrace [17] was used. Ftrace is typically a function tracer that examines the occurrence between disabled and enabled interrupts and schedules the preemption task. The lists of real-time related functions are presented in Figure 10. These functions can be featured as indirect calls or affected mode switches. Considering that most real-time functions call update_curr_dl() functions, they are the most essential functions among them. The caller functions of update_curr_dl() are usually task_tick_dl() and __dequeue_dl_entity(). The indirect functions are not directly affected by MC patches but are sometimes affected by updating the virtual deadlines (VD). The VD performs a constrained deadline in Linux scheduler implementations, which is considered to be complicated both theoretically and in terms of implementation.

In Figure 10, the results of ftrace on real-time functions are presented. The test run at every mode-switch instance also represents the scheduling overload. The overhead of an essential function update_curr_dl() shows that all implemented MC schedulers have a mean overhead of more than 143%. The scheduling overhead level can be considered critical; however, the least frequency call can be defined by the function task_tick_dl(), which is set to 1000 Hz (1 ms cyclic), and this frequency is the maximum in Raspberry Pi 3B+ configurations. By referencing the frequency of scheduling overheads, the worst-case effect on the entire system can be derived as:
(10)$$\frac{overhead\;(update\_curr\_dl())}{scheduling\;cycle}=\frac{1.908\;\mu s}{1\;\mathrm{ms}}=0.19\%,$$
(11)$$\frac{overhead\;(task\_tick\_dl())}{scheduling\;cycle}=\frac{3.971\;\mu s}{1\;\mathrm{ms}}=0.39\%.$$

From the calculations, we conclude that a scheduling overhead of less than 1% hardly violates the system. For the DMR test, the MC schedulers are already proven to be functional for real-time guarantees.

The runtime scheduling overhead for each scheduler is displayed in Figure 11. It can be observed that the runtime phase overheads of system-level and task-level MC schedulers are twice that of the vanilla EDF scheduler as observed in MC lists, based on structure type. However, the additional structure is immune to the number of tasks because referencing each scheduling entity has a cost of O(1) for each function.

#### DMC Patch Overhead

In previous experiments, it was observed that the DMC module was independent and did not significantly affect the scheduler. The effect of each real-time function on the MC scheduler was 2.15% at the system level and 7.39% at the task level. This is because of the independent modular design of the DMC features of the scheduler.

## 7. Related Works

From the perspective of task scheduling, the integration of different levels of safety components can be handled by MC scheduling, which was first proposed by Vestal [6]. Recent trends towards gaining more schedulability for LC tasks in MC systems are adaptive scheduling [8,9] and multiprocessing [18,19,20] approaches. Adaptive schedulers include EDF-based [9] and RM-based [8] schedulers. Our approach focuses on EDF-based schedulers, the proofs, and implementations. Multiprocessing solutions include partitioned scheduling and simplex architecture scheduling, and these solutions have been adapted to actual products by manufacturers because of the need for MC domains [21]. A summary of previous studies related to our approach is presented in Table 6 for varying scheduling domains, LC task considerations, implementation, and costs. To the best of our knowledge, our approach is the first to implement extended schedulable features for LC tasks in the Linux Kernel, for distributed MC systems.

In this section, we describe in detail how our approach differs under the following considerations. First, the selected works vary in target scope. Hot-patching [22,23] provides seamless updates on tasks at runtime, but the task should stay in a single system that is not a direct solution for scheduling overheads. Redundancy [24,25], multiprocessing [18,19,20], and heterogeneous global scheduling [26] comprise hardware configurations and are mainly targeted to perform a singular concrete system, which may cause a single point of failure (SPOF) in system failure. Mobile agent- [27,28,29,30] and communication-based [31,32] works are helpful in the scope of distributed systems, but the objective is not task preservation. Second, some works must consider shared resources during system design. Hot-patching, redundancy, and multiprocessing are bound to a singular system or exist to compose a single system, which results in a robust singular system; however, these solutions have the same memory-sharing requirement, which may cause the single point of failure in system failures. Third, the cost of each solution is important in the decision stage of production. In this context, hardware-based solutions such as redundancy and multiprocessing are considered to be expensive compared with other software approaches. Finally, considering the worst-case schedulability of all MC tasks is critical in safety-critical systems. In the domain of MC policies and some important LC tasks considerations, adaptive MC scheduling is regarded as the optimal solution among the selected works. In other words, in the case of scheduling overheads, using the adaptive MC scheduling enables maximum MC, HC, and LC tasks to be accepted. This acceptance is quantified as the schedulability or acceptance of a given task system. Beyond adaptive MC, our approach extends the schedulability by supplementing the dismissed tasks of the scheduler and achieving a greater probability of accepting the task systems. Among the comparable works, our approach enables the achievement of a higher acceptance of tasks and practical usage in Linux Kernel while saving hardware costs.

## 8. Discussion

With the constraint that task migration latency is relatively negligible in any MC task utilization, using DMC with the EDF-based MC schedulers offers overwhelmingly improved important LC task schedulability. For the EDF-VD scheduler, the deadline miss ratio of LC tasks improved by 25.53% from 0.47% to 0.35%, and for the MC-ADAPT scheduler, the improvement in the deadline miss ratio was 100%, from 0.12% to 0%. The proposed method’s approach of gaining more CPU time also worked well in the test case. The results showed that the schedulability of our DMC scheduler and task migration was predictable in the case of a mode switch. There are, however, some limitations to our approach.

Real-Time guarantee on scheduler: To guarantee real-time assurance for both the tasks and scheduler, the latter was designed to always have the earliest deadline among tasks to be scheduled on a first-priority basis.Scheduling overhead: For the Linux scheduler function measurement, the MC scheduling had up to 2.3 times scheduling overhead; however, it did not violate the schedulability of the task system and remained feasible. As future work, we intend to arrange the HC-WCET of the HC task adaptively by measuring the system runtime to lower the total utilization and scheduling overhead; however, the focus of this study was on the worst-case schedulability under a fixed HC-WCET of tasks.Preconditions: The time or task migration should be smaller than the WCET of the important LC task and should be schedulable in the HC-mode source system. The destination system must be able to schedule migrated tasks, and there is a constraint that only one task can be migrated at a time.Hardware dependency: Task migration comprises network transfers, and the success of task migration mainly depends on the hardware devices that determine the speed of communications. In Section 6.3, we introduced the latency of task migration, which is mainly concerned with the hardware connection type and bandwidth reservations. To bind task migration in hard real-time, a predictable hardware design should be discussed for stabilizing the network latency.Dual-criticality consideration: As mentioned in EDF-VD theory, the dual-criticality model is easily generalized to more than two criticality levels [10]. In this context, there are two motivations for considering two levels of criticalities. The first is explicitness, which provides easy and sufficient understanding for the critical instant of MC scheduling. The second motivation is the implication of proving more than two criticality levels. For example, in automotive systems, there are five safety-criticality levels, which can be denoted in the ascending order of criticality as A, B, C, D, and E (E is the highest criticality). Considering that critical instants rarely occur simultaneously, we can categorize the criticality groups at critical instants in two levels (in most studies on MC, two criticalities are defined as HI and LO). For example, suppose a function having criticality level C had the critical instant exceeding its initial worst-case execution time. In this situation, lower criticality levels (A and B) can be considered as LO criticality levels (conversely, C, D, and E as HI levels).Certification of the overall distributed system: Certification is an important aspect of safety-criticality systems representing the *reliability* of target systems; however, in distributed systems, a higher-level certification is required for an overall distributed system rather than considering every single system. The certification is measured as a worst-case upper bound; therefore, the overall certification should have a higher rate if one of the systems requires higher certifications. This results in the relationship of certification between each system and the overall distributed system as disjunction in logics as denoted in Table 7. The LC task $\tau_{3}$ requires a higher overall certification than the certification of system A. More than two systems complicate the overall certification process.

## 9. Conclusions

The resiliency of LC tasks was evaluated from the perspective of schedulability. The feasibility of task migration in the MC scheduler was verified by offline and online schedulability analyses. The results showed that task migration was feasible using the EDF-based MC scheduler. An MC scheduler with task migration features was implemented on the SCHED_DEADLINE Linux Kernel. Compared with conventional MC schedulers, the deadline miss ratio decreased significantly for the task-level MC scheduler without violating the schedulability of the HC tasks. The results of the deadline test showed that the compatibility with the task-level MC scheduler was higher than that of the system-level MC scheduler, where the task-level had O(1), deadline miss ratios compared with O(n) for the system-level on *n* tasks. In addition, based on the violation test on HC tasks, the distributed approach proposed in this study was more stable on system-level MC schedulers. In conclusion, under the condition that the HC task ratio has lower than 47% of overall task systems with 80% of total utilization, the task-level approach with task migration has extensively higher sustainability on LC tasks. In the future, our work will be applied to more detailed and realistic mixed-criticality case studies, such as truck platooning, autonomous/connected cars, and aerial/aerospace systems.

## Figures and Tables

**Figure 1 sensors-22-01926-f001:**
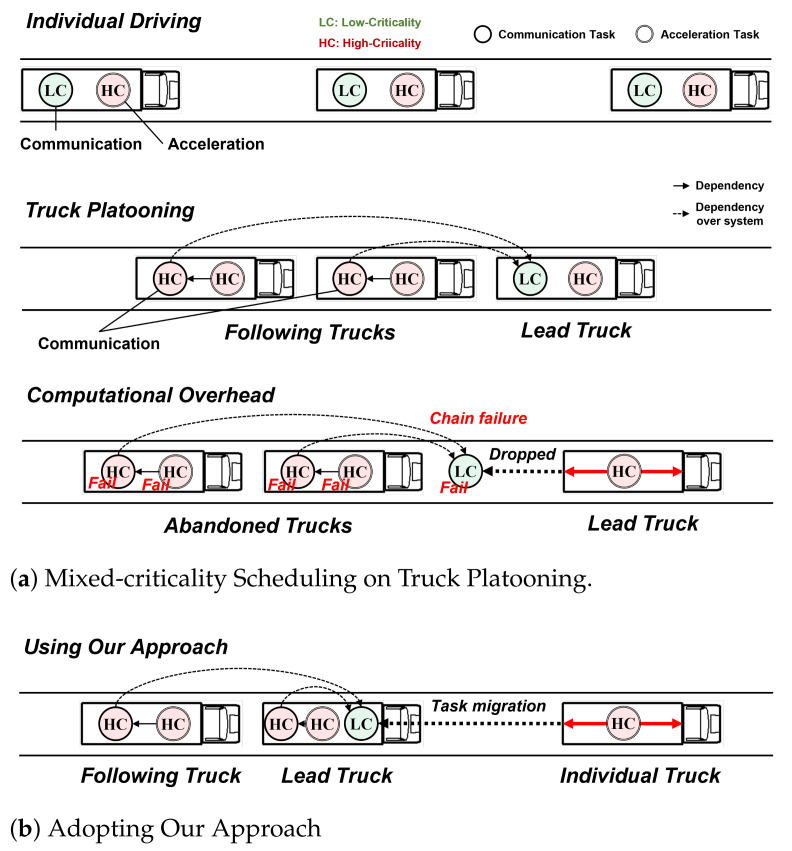
Overcoming the Chain Failure of Mixed-criticality Scheduler by Adopting our Approach.

**Figure 2 sensors-22-01926-f002:**
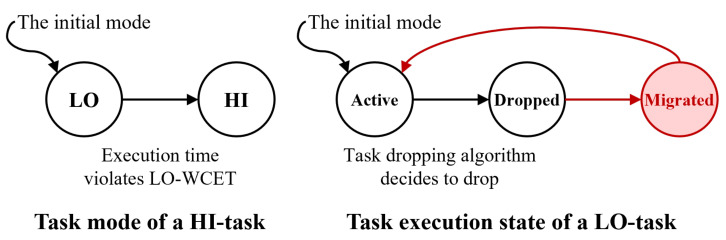
Extended behavioral model of tasks.

**Figure 3 sensors-22-01926-f003:**
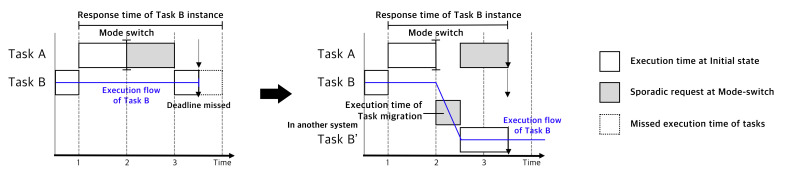
Task Migration at Mode switch.

**Figure 4 sensors-22-01926-f004:**
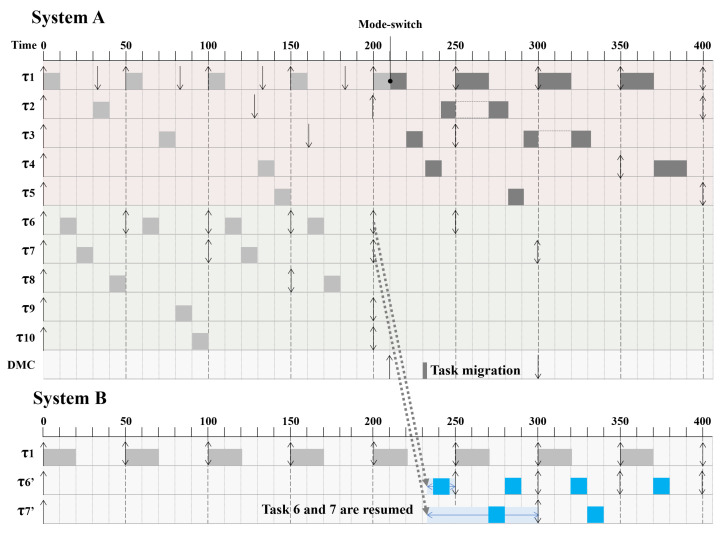
Gantt chart of mode-switch.

**Figure 5 sensors-22-01926-f005:**
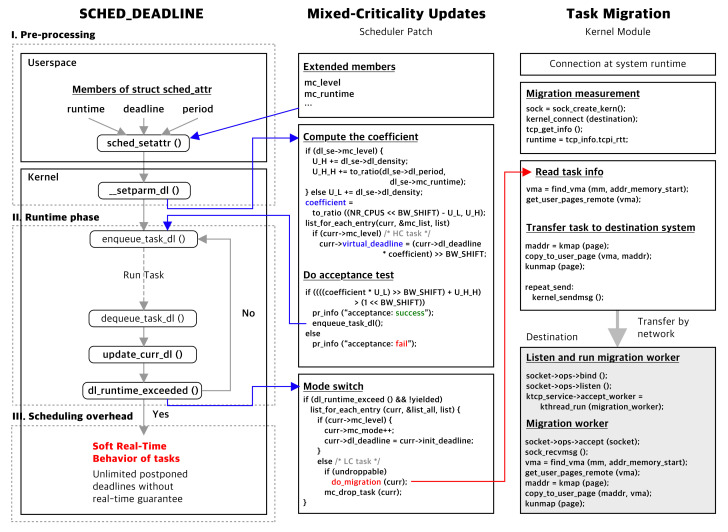
Structure of our distributed system-level approach.

**Figure 6 sensors-22-01926-f006:**
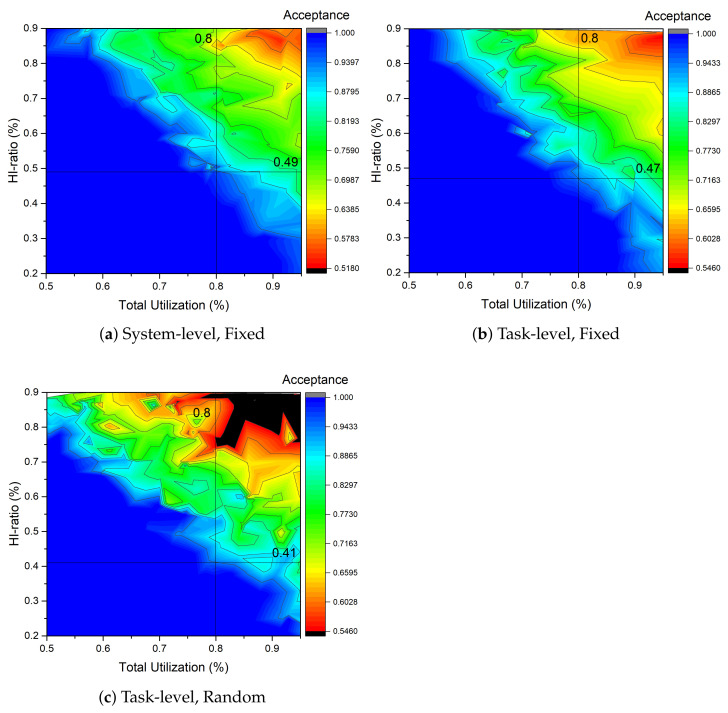
Distribution of acceptance test on (**a**) system-level MC scheduler (EDF-VD) and (**b**,**c**) task-level MC scheduler (MC-ADAPT) with fixed (2.0 times) and random (average 2.0 times) replenishment of low criticality runtime. The result with the (**c**) random task-level MC scheduler has approximately 0.4 maximum HC ratio on 0.8 total utilization, which is approximately 20% of the HC ratio loss compared to the (**a**,**b**) fixed conditions having 0.5 HC ratio acceptance. In the figures, total utilization starts at 0.3, and an acceptance less than 1.0 is considered as failure.

**Figure 7 sensors-22-01926-f007:**
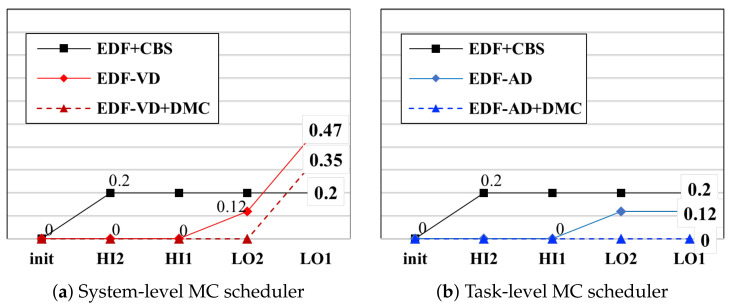
Accumulative result of mode-switch-triggered DMR test on (**a**) system-level and (**b**) task-level MC schedulers under task-level scheduling order.

**Figure 8 sensors-22-01926-f008:**
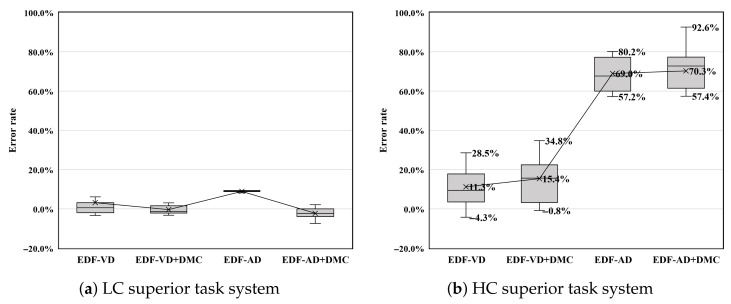
Violation test result of LC mode HC tasks with mode-switch triggered deadline test on (**a**) LC superior task system (HC:LC = 35:47) and (**b**) HC superior task system (HC:LC = 47:35).

**Figure 9 sensors-22-01926-f009:**
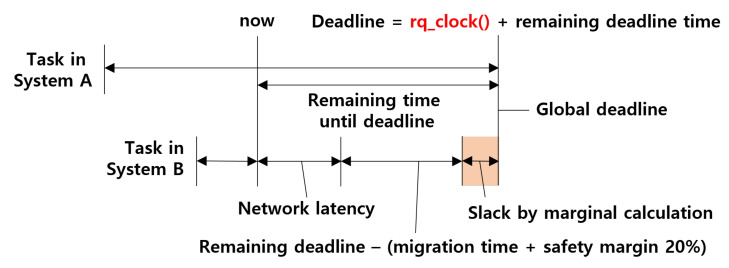
Transformation into global deadline.

**Figure 10 sensors-22-01926-f010:**
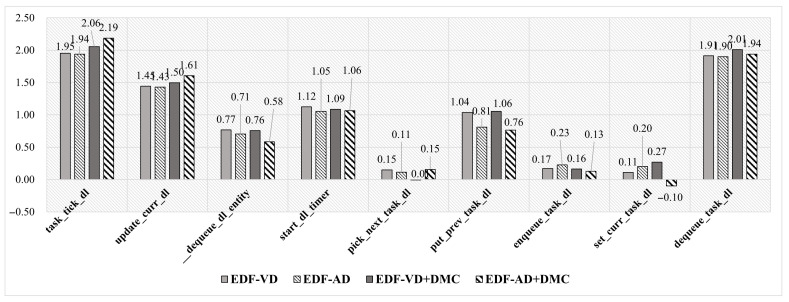
Ftrace: scheduling overhead comparison in graphs.

**Figure 11 sensors-22-01926-f011:**
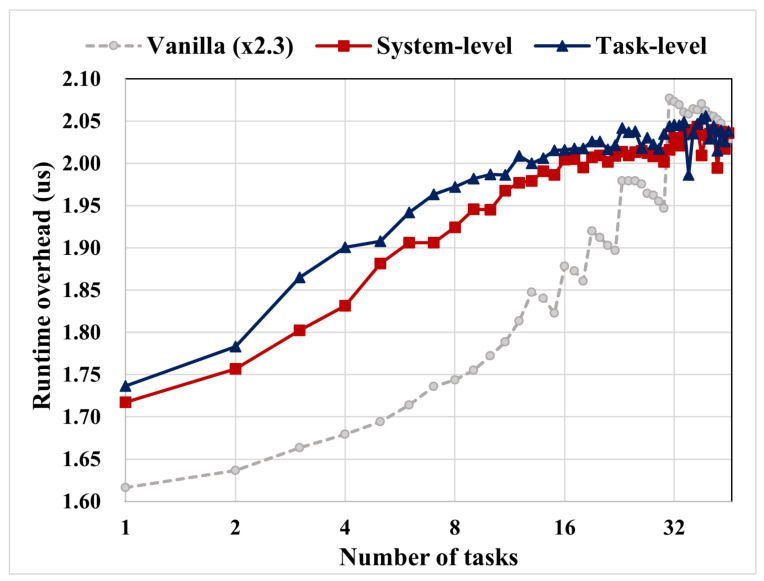
Runtime scheduling overhead by number of tasks.

**Table 1 sensors-22-01926-t001:** Task model for MC schedulers.

Symbol	Definition
$\chi_{i}$	Criticality level of task *i*, $L_{i}\in \left\{L,H\right\}$.
$C_{i}^{L}$	Low-Confidence WCET (LC-WCET).
$C_{i}^{H}$	High-Confidence WCET (HC-WCET).
$D_{i}$	Relative Deadline of task *i*.
$T_{i}$	Period of task *i*.

**Table 2 sensors-22-01926-t002:** MC Task System for DMC (time unit: ms).

Task	$\chi_{i}$	$C_{L}$	$C_{H}$	T/D	$U_{\chi_{i}}^{L}$	$U_{\chi_{i}}^{H}$
$\tau_{1}$	HC	10	20	50	20.00%	40.00%
$\tau_{2}$	HC	10	20	200	5.00%	10.00%
$\tau_{3}$	HC	10	20	250	4.00%	8.00%
$\tau_{4}$	HC	10	20	350	2.86%	5.71%
$\tau_{5}$	HC	10	20	400	2.50%	5.00%
$\tau_{6}$	LC	10	0	50	20.00%	0%
$\tau_{7}$	LC	10	0	100	10.00%	0%
$\tau_{8}$	LC	10	0	150	6.67%	0%
$\tau_{9}$	LC	10	0	200	5.00%	0%
$\tau_{10}$	LC	10	0	200	5.00%	0%
$\tau_{m}\left(V\right)$	-	-	1	81	-	1.2%
$\tau_{m}\left(A\right)$	-	-	1	16	-	6.1%

**Table 3 sensors-22-01926-t003:** Time and space complexity of Algorithm 1.

EDF-VD System-Level	Time Complexity	Operation Cost
Preprocessing phase	$O(n^{2})$	Cost of each task *i*: 8i+13(i=1…N) Cost of all tasks: $8\frac{N(N-1)}{2}+13N$ $\to 4N^{2}+9N$
Runtime: Initial state	O(1)	4
Runtime: Mode-switch	O(n)	6N+1

**Table 4 sensors-22-01926-t004:** Time and space complexity of Algorithm 2.

MC-ADAPT Task-Level	Time Complexity	Operation Cost
Preprocessing phase	$O(n^{2})$	Cost of each task *i*: 20i+14(i=1…N) Cost of all tasks: $20\frac{N(N-1)}{2}+14N$ $\to 10N^{2}+4N$
Runtime: Initial state	O(1)	4 (Same as EDF-VD)
Runtime: Mode-switch	O(n)	5N+12 (Pseudo-polynomial)

**Table 5 sensors-22-01926-t005:** Network latency on different connections.

Network Settings	Avg. (Stdev.) (ms)
Wi-Fi router	25.83 (±29.43)
Wi-Fi-direct	3.43 (±2.66)
Wi-Fi-direct+module	1.94 (±0.35)

**Table 6 sensors-22-01926-t006:** Comparison of a selection of previous studies.

Prior Works	Scope of the Solution	Shared Resource Requirements	Mixed-Criticality Considered	Low-Criticality Tasks Considered	System Implementation and Evaluation	Cost-Effective
Hot patching [22,23]	A task bound in a single system	Yes, shared memory	No	No	Yes	Yes
Mobile agent [27,28,29,30,33]	Portable program in Distributed system	No	No	No	Yes	Yes
Redundancy [24,25]	Multiple systems for a single purpose	Yes, redundant system and connections for mission take-over	No	No	Yes	No, because of the additional system costs
Communication [31,32]	Portable data in distributed system	No, using existing communications	Yes, by applying MC policies in network protocol	No	No, only in network route	Yes
Task migration [34]	Portable task in distributed system	No	Yes	No	No, mainly theoretical proofs	Yes
Heterogeneous Global Scheduling [26]	Overall quality in heterogeneous distributed system	Yes	No	Overall QoS	ATmega328p and ATmega2560	Yes
Multiprocessing [18,19,20]	Multi-core for a single system	Yes, shared memory	Yes	No	No, mainly theoretical proofs	No, because of the additional processing units
Adaptive MC [8,9]	All MC-tasks in a single system	Yes, shared memory and CPU	Yes	Yes	No, mainly theoretical proofs	Yes
Elastic MC [35]	All MC-tasks in a single system	Yes, shared memory and CPU	Yes	Overall QoS of LC-tasks	No, mainly theoretical proofs	Yes
Our approach (DMC)	All MC-tasks in distributed system	No	Yes	Yes	Yes	Yes

**Table 7 sensors-22-01926-t007:** Certification of overall distributed system.

Certification on Tasks	Each Systems		Overall System
	System A	System B	
CERT($\tau_{1}$)	HI	HI	HI
CERT($\tau_{2}$)	HI	LO	HI
CERT($\tau_{3}$)	LO	HI	HI
CERT($\tau_{4}$)	LO	LO	LO

## Data Availability

Not applicable.

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
