# Peer review of "Task Migration and Scheduler for Mixed-Criticality Systems"

_sensors, 2022, doi:10.3390/s22051926_

Round 1
Reviewer 1 Report
The article “Task Migration and Scheduler for Mixed-Criticality Systems” targets the field of Mixed Criticality Systems on distributed systems. The paper provides a theoretical and a practical solution for scheduling mixed criticality tasks on such systems. Feasibility tests are also provided. The algorithm is analyzed and compared in terms of feasibility, scheduling overhead and response time to other state of the art algorithms like EDF-VD.
The article is well written and the fact that it also provides a reference implementation on Linux brings the research closer to the industry.
This reviewer suggests some minor modifications:
- The three dot operators from equation (2) could be specified in the text or replaced with “therefore” and “because” for a better clarity
- More optimal - Line 334-335 – should be replaced with “more efficient”
- Figure ?? line 525 – the number of the figure is missing
- Section?? Line 617 – the number of the section is missing
Reviewer 2 Report
This work fits better in a journal such as Real-Time Systems or related. Most of the referenced papers are from this area. Why are you submitting this manuscript to this journal? You could submit it to a journal like this. Technically, it is a good paper and might be more suitable for other journals.
a set of HC tasks is denoted --> a set of HC tasks are denoted
a set of LC tasks is denoted --> a set of LC tasks are denoted
Based on IEC 61508 or ISO 26262? page. 6
Use the equation for all equations or migrate the paper to Overleaf / Latex
Expand better Table 3 and Table 4
Improve vision of figure 6
Reviewer 3 Report
This paper focuses on mixed-criticality systems for the high-criticality (HC) tasks and low-criticality (LC) tasks. The authors propose a safety-guaranteed and inexpensive scheduling for distributed mixed-criticality (DMC) scheduling which is a platform for scheduling LC tasks dropped in distributed systems in theory and in practice. When the HC task requires more than the given execution time, the HC task exceeds the runtime, and some LC tasks are sacrificed in conventional MC schedulers. The results show that the approach extended the schedulability of LC tasks without violating the HC tasks.
- This paper has 25 pages; please explain why some of the content needs to be mentioned, e.g.,6.0.1. Figure 6, Figure 10, Figure 11.
- In evaluation, please add more details for the experimental observations, e.g, Figure 7, Figure 10 & 11 (what these values mean, why drop at the number of tasks 3X).
- The paper has some typos and please re-check it again, e.g., the symbol “?”.
- In 4.1.3, “Utilization is the CPU usage percentage, but the calculation must be done with the actual execution time guaranteed at the instant of the mode switch.” Seems the analysis depends on the example, so why did the authors choose the example task system in Table 2? If changing different examples, is it possible to get different results?
